# Simple and Precise Description of the Transformation Kinetics and Final Structure of Dual Phase Steels

**DOI:** 10.3390/ma14071781

**Published:** 2021-04-04

**Authors:** Jan Kohout

**Affiliations:** Department of Mathematics and Physics, Faculty of Military Technology, University of Defence, Kounicova 65, CZ-662 10 Brno, Czech Republic; jan.kohout@unob.cz; Tel.: +420-973-443-283

**Keywords:** dual phase steel, austenitisation, JMAK equation, equation of autoinhibition, Arrhenius equation, predictive curves

## Abstract

The kinetics of diffusion-dependent phase transformations (including austenitisation of ferrite in dual steels or ferritic nodular cast irons) is very often described by the Johnson–Mehl–Avrami–Kolmogorov (JMAK) equation. This description is not complete when the conversion is only partial due to insufficient overheating, as the equilibrium fraction of ferrite transformed into austenite cannot be determined directly from the JMAK equation. Experimental kinetic curves of partial austenitisation at various temperatures can be fitted using the JMAK equation, but the equilibrium fraction of the newly formed phase for each temperature has to be calculated as a regression parameter. In addition, the temperature dependence of the kinetic exponent in the JMAK equation is quite complicated and cannot be expressed by a simple general function. On the contrary, the equation of autoinhibition used for the description of austenitisation kinetics in present work directly gives the equilibrium fraction at partial conversion. It describes transformation kinetics at various temperatures independently of whether the conversion is complete or partial. Rate constants of the equation of autoinhibition depend on temperature according to the Arrhenius equation. In addition, the equation of autoinhibition has no weakness as the JMAK equation has, which consists in questionable temperature dependence of kinetic exponent.

## 1. Introduction

Conventional dual phase steels containing soft ferrite and hard martensite are high strength steels that are widely applicable in industrial production. The high strength of these materials allows reduction in the weight of structures, leading to energy savings, decreasing total cost and increasing utility value of products. In addition, dual steels play a special role in the automotive industry, due in large part to their excellent crash performance [1], and in the armament industry, as their favourable ballistic performance can be exploited for light armour production [2]. The production and mechanical properties of dual steels were studied in depth many years ago, e.g., by Cairns and Charles [3,4], Tamura et al. [5] and Davies [6,7]. Some researchers paid special attention to the martensite content in dual steels [8,9], or showed that martensite particle size [10] and morphology [11,12] also play important roles. The contributions of ferrite and martensite to the overall mechanical behaviour of dual steel were described using the modified law of mixtures [8], or the neural network model [13].

Heat treatment of dual steels leads to partial austenitisation, and during cooling austenite transforms into martensite. The kinetics of austenite formation at given temperature is mostly described by the Johnson–Mehl–Avrami–Kolmogorov (JMAK) equation [14,15,16,17,18,19,20], including recent papers specifically on dual steels [21,22,23], only rarely it is replaced by modelling and simulations, e.g., [24]. Usual form of the JMAK equation is:(1)$$p(t)=1-\exp(-k\;t^{n}),$$
where *p* is conversion, i.e., the relative content of the newly formed phase. Theoretical considerations lead to integer values of kinetic exponent *n* between 1 and 4 depending on nucleation types of the new phase (homogeneous or heterogeneous) and on the shape of nuclei (spherical, planar or fibrous). Some other considerations taking into account the subsequent growth of nuclei after their mutual contact lead to half-integer values of parameter *n* 0.5, 1.5, or 2.5. Regression of the measured kinetic curves leads to various values of *n* without any tendency to approach the above values. It is often explained by the fact that each stage of the phase transformation needs its own value of parameter *n* and its value from regression of the whole kinetic curve is a certain representative average of values corresponding to different stages of the phase change. Very illustrative demonstration of more stages of phase transformation described by the JMAK equation with individual values *n* is given in [20] (see Figures 10–12 in [20]) where the fit ln *t*−ln ln [1/(1 − *p*)] linearizing the JMAK equation is used. In this fit the whole kinetic curves are presented by two or three linear parts with individual slopes representing individual values of kinetic exponent *n*.

To describe austenitisation at various temperatures, the temperature dependence of the rate constant *k* can be described by the Arrhenius equation:(2)$$k(T)=k_{\infty }\exp\left(-\frac{\varepsilon }{R\;T}\right),$$
where *ε* is the activation energy of the transformation under study, *R* is the universal gas constant and *T* is the absolute temperature of isothermal austenitisation (in all equations the temperature is expressed in kelvins). The pre-exponential factor *k*_∞_ expresses the rate constant (kinetic coefficient) *k* at infinite temperature, *k*_∞_ = *k*(*T*→∞). Also, the kinetic exponent *n* in the JMAK equation is usually strongly temperature-dependent, but this mostly non-monotonic dependence cannot be generally expressed by a simple function.

Phase transformation is a rather complicated process and, therefore, its simple kinetic description must focus only on one, or a few, crucial stages or influences. The JMAK equation focuses mainly on the nucleation stage of new phase. In addition, a difference in specific volumes of original and new phase can play a substantial role, resulting in suppressing phase conversion. In simple description, this results in a delay in the conversion kinetics, which is proportional to the phase fractions of original phase 1 − *p* and new phase *p*, i.e., to the product of those fractions *p*(1 − *p*). If the phase conversion without the influence of different specific volumes is described by the simplest kinetic equation (for first-order reaction), then:(3)$$\frac{dp}{dt}=k(1-p)-k_{s}p(1-p)\;i.e.,\;p(t)=1-\frac{k-k_{s}}{k\exp[(k-k_{s})t]-k_{s}},$$
where the rate constant *k*_s_ describes the influence of different specific volumes and its minus sign means suppressing phase conversion. In addition, here the temperature dependence of the rate constants *k* and *k*_s_ can be described by the Arrhenius Equation (2) with corresponding activation energies *ε* and *ε*_s_, respectively.

As Equation (3) describes certain deceleration of phase conversion due to different specific volumes of original and new phase, it can be called equation of autoinhibition. This equation describes transformation kinetics with accuracy comparable to that of the JMAK equation, removes the problem of the temperature dependence of the kinetic exponent *n*, and can directly describe partial conversion when the overheating is not sufficient for complete conversion, as shown in the previous paper [19]. The aim of the current paper is to apply the equation of autoinhibition to the description of kinetics of austenite formation in dual phase steels and to introduce certain modification for the case, when dual steel contains only little part of martensite.

## 2. Experimental

The austenitisation kinetics was studied using the experimental dependence of the content of the transformed phase on time and temperature as published by Asadabad et al. [17]. Steel with a composition of 0.11 wt.% C, 0.14 wt.% Si, 1.30 wt.% Mn, 0.45 wt.% Cr and 0.03 wt.% Al was studied. The 200 × 40 × 16-mm^3^ slabs were cast in sand mould in the vertical position, hot rolled at 1200 °C to 1.8-mm thickness, homogenised at 950 °C for 1 h and normalised at 900 °C for 15 min. The obtained sheets were austenitised in an electric resistance furnace at constant temperatures of 730 °C, 760 °C, 790 °C, 820 °C, and 850 °C with holding times between 5 and 1800 s and then quenched in water. The austenite ratio was expressed as the martensite volume fraction determined by point counting method, according to ASTM E562-83 standard [25], using optical and scanning electron microscopy. More details can be found in the original paper [17]. Experimental results are presented in Figure 1 (see only experimental points).

## 3. Regression Using the JMAK Equation

Figure 1 also presents a regression of the experimental results [17] using the JMAK equation (see fitting lines). As in the original paper [17], individual values of the rate constant *k* and the kinetic exponent *n* are considered for each experimental temperature. Here the following regression function is used:(4)$$p(t)=p_{\mathrm{eq}}[1-\exp(-k\;t^{n})],$$
where *p*_eq_ is the equilibrium conversion reached after a sufficiently long time (equal to 1 for complete conversion and less than 1 for partial conversion), while the authors of the paper [17] used the same equation in linearised form:(5)$$\ln\ln[1/(1-p/p_{\mathrm{eq}})]=n\ln t+\ln k.$$

Resulting values of regression parameters *k* and *n* obtained by nonlinear regression (present work) differ considerably from the values obtained using linearized procedure (paper [17]), see Table 1, while the values of equilibrium conversion *p*_eq_ are the same in both fits. Nevertheless, the regression curves are hardly distinguishable (cf. Figures 1 and 8 in paper [17]).

## 4. Regression Using the Equation of Autoinhibition

Figure 2 presents a regression of the experimental results of Asadabad et al. [17] using the equation of autoinhibition (3) together with the Arrhenius equation (Equation (2)) used for both *k* and *k*_s_ rate constants. The family of curves is determined by only four parameters: *k*_∞_, *k*_s__∞_, *ε*, and *ε*_s_. Although this fit is noticeably worse than that in Figure 1 obtained using the JMAK equation (comparison of both fits is presented in Figure 3), it can be considered to be very successful: the fit using the JMAK equation requires not 4, but 14 regression parameters (*k*, *n* and *p*_eq_ for each of five temperatures with the exception of *p*_eq_, which is equal to 1 for 850 °C when the conversion is complete). Nevertheless, the biggest advantage of the equation of autoinhibition is that there is no need for a separate equilibrium conversion parameter *p*_eq_ for each temperature, but equilibrium conversions for all temperatures follow directly from the equation of autoinhibition.

Figure 1, Figure 2 and Figure 3 show that complete conversion is reached at 850 °C, while it is only partial at 820 °C and at lower temperatures. This means that the minimum temperature *T*_min_ at which complete conversion is reached after a sufficiently long time (theoretically infinite time) lies between these two temperatures. If the equation of autoinhibition is used to describe the conversion, the relationship for this temperature can be simply derived [19]:(6)$$T_{\min}=\frac{\varepsilon -\varepsilon_{s}}{R\;\ln\frac{k_{\infty }}{k_{s\;\infty }}}.$$

For this temperature, the rate constants *k* and *k*_s_ have the same value, i.e., *k*(*T*_min_) = *k*_s_(*T*_min_) = *K*, which can be expressed in terms of the quartet of parameters *k*_∞_, *k*_s__∞_, *ε*, and *ε*_s_ as:(7)$$K=k_{s\;\infty }{}^{\frac{\varepsilon }{\varepsilon -\varepsilon_{s}}}/k_{\infty }{}^{\frac{\varepsilon_{s}}{\varepsilon -\varepsilon_{s}}}.$$

Then, instead of the quartet of parameters *k*_∞_, *k*_s__∞_, *ε*, and *ε*_s_, the quartet of parameters *T*_min_, *K*, *ε*, and *ε*_s_ can be used and the Arrhenius equation for rate constants *k*(*T*) and *k*_s_(*T*) takes the form [19]:(8a)$$\begin{array}{c} k(T)=K\exp\left[-\frac{\varepsilon }{R}\left(\frac{1}{T}-\frac{1}{T_{\min}}\right)\right] \end{array}$$
(8b)$$\begin{array}{c} k_{s}(T)=K\exp\left[-\frac{\varepsilon_{s}}{R}\left(\frac{1}{T}-\frac{1}{T_{\min}}\right)\right] \end{array}.$$

Then the limit temperature *T*_min_ can be calculated not only using Equation (6), but also directly as regression parameter when regression function (3) is supplemented by the Arrhenius equation in the form of (8). Its value *T*_min_ = 835.3 °C slightly differs from the temperature *Ac*_3_ = 849 °C calculated by Asadabad et al. [17] using the phenomenological equation for *Ac*_3_ published by Andrews [26]. Their difference is simply explicable: value 849 °C is valid for dwells usually used in thermal treatment, while value 835 °C is a limit for infinite time.

## 5. Construction of Predictive Curves Using the Equation of Autoinhibition

Conversion *p*(*t,T*) expressed as a function of time and temperature using the equation of autoinhibition (3) and the Arrhenius equation, e.g., in the forms of (2) or (8), allows the regression of experimentally measured kinetic curves giving the values of the quartet of regression parameters. For the results of Asadabad et al. [17], the values *T*_min_ = 835.3 °C, *K* = 0.06864 s^−1^, *ε* = 279.6 kJ/mol and ε_s_ = 169.6 kJ/mol were obtained. The values of this parameter quartet fully describe the austenite formation of the studied material within the model of autoinhibition described here, therefore, substituting them into Equations (3) and (8) yields the family of predictive curves for temperatures within and near the range of test temperatures. Starting from 900 °C downwards, Figure 4 presents predictive curves in temperature intervals of 20 °C (as well as a predictive curve for *T*_min_ = 835.3 °C). Unfortunately, the validity of the model is limited by the temperature *Ac*_1_, below which austenite is not formed. The value of this temperature calculated by Asadabad et al. [17] using the phenomenological equation published by Andrews [26] is *Ac*_1_ = 721 °C. Therefore, the predictive curve for this temperature (drawn in Figure 4 as a dotted line) is not valid and must be replaced by the line expressing *p* = 0 (the solid line below the arrows in Figure 4).

## 6. Modification of the Arrhenius Equation Near Temperature *Ac*_1_

Figure 4 shows apparently that the prediction can be used only for austenitisation temperatures higher than 730 °C, i.e., for martensite volume fraction higher than 0.2 = 20%. The cause consists in the fact that rate constant *k* describing austenite formation (see analogy to Equations (2) and (8a)):(9)$$k(T)=k_{\infty }\exp\;\left(-\frac{\varepsilon }{R\;T}\right)=K\exp\left[-\frac{\varepsilon }{R}\left(\frac{1}{T}-\frac{1}{T_{\min}}\right)\right]\;$$
is nonzero for all temperatures (excluding absolute zero), while austenite formation can occur only at temperatures higher than 721 °C. Step change:(10)$$\begin{array}{cc} k(T)=0 & \mathrm{for}\;T\leq 994\;K\;(721\;{}^{\circ}C)\\ k(T)=K\exp\left[-\frac{\varepsilon }{R}\left(\frac{1}{T}-\frac{1}{T_{\min}}\right)\right]\; & \mathrm{for}\;T\;>\;994\;K\;(721\;{}^{\circ}C) \end{array}$$
solves the problem only qualitatively, but transient change using, e.g., tanh function:(11)$$\begin{array}{cc} k(T)=0 & \mathrm{for}\;\;T\leq 994\;K\;(721\;{}^{\circ}C)\\ k(T)=K\exp\left[-\frac{\varepsilon }{R}\left(\frac{1}{T}-\frac{1}{T_{\min}}\right)\right]\;\tanh[a\;(T-994)] & \mathrm{for}\;T\;>\;994\;K\;(721\;{}^{\circ}C) \end{array}$$
where *a* is the slope of the transition, leads to better fit and especially to better predictive curves near above 721 °C. This slope *a* is then the fifth regression parameter.

Result of regression using Equation (11) instead of (8a) or (2) for rate constant *k*(*T*) is shown in Figure 5. This regression is better than that in Figure 2, although the sum of squares decreases only by 6.2% (from 0.0384 to 0.0360). Figure 6 compares regressions from Figure 2 and Figure 5 differing in functions describing rate constant *k*(*T*) by Equation (2) or (8a) (dashed lines) and (11) (full lines). The comparison shows that better regression is obtained namely for temperatures of 730, 760 and also 790 °C. Using resulting values of regression parameters *T*_min_ = 836.8 °C, *K* = 0.07198 s^−1^, *ε* = 279.1 kJ/mol, *ε*_s_ = 175.2 kJ/mol, and *a* = 0.1658 K^−1^, the family of predictive curves was drawn, see Figure 7, where the problem with temperatures close above 721 °C disappeared.

## 7. Discussion

In spite of its simplicity, the JMAK equation is sophisticated and universal because it can describe practically all types of diffusion-based phase change in all stages, including the formation of the nuclei of the new phase, the independent growth of those nuclei, and the growth when their surfaces reach one another. Different mechanisms of growth predominate in different stages, for which different values of kinetic exponent *n* can be considered [20]. Sinha et al. [27] took this fact into account by considering the temporal dependence of this exponent. All experimental results show that this exponent depends on temperature, and Rios [28] took this dependence into account in his theoretical considerations. In fact, both the temporal and temperature dependence of the kinetic exponent should be considered, but then the JMAK equation loses its basic advantage of simplicity.

While the JMAK equation can be adapted for individual stages of austenite formation (with individual values of kinetic exponent *n*), the equation of autoinhibition describes austenite formation (and many other processes) on a very general level, taking only two basic driving forces into account: overheating acting in favour of forming a new phase, and the newly formed phase acting against the continuing conversion (due to factors such as different specific volumes of origin and the new phase, or also chemical liquation of alloying additions). The advantage of the equation of autoinhibition is that it is able to describe the final equilibrium of both these driving forces at any given temperature (including the equilibrium fraction of the newly formed phase), as well as the temporal course of arriving at equilibrium (i.e., the kinetics of conversion). The JMAK equation is not able to describe the equilibrium fraction, but if it is known, the JMAK equation describes the kinetics in many cases even better than the equation of autoinhibition: (i) the JMAK equation can describe the kinetics of a nearly arbitrary phase change (based on diffusion), if the initial and final states are known (by choosing a suitable value for the kinetic exponent); (ii) the equation of autoinhibition is able to determine that final state and, to describe the temperature-dependent equilibrium (together with the kinetics of its formation) of two nearly arbitrary counteracting driving forces with different temperature dependences (given by different values of activation energy *ε* and *ε*_s_). This makes the equation of autoinhibition a very strong tool for the description (or more strictly speaking, for the modelling) of a wide variety of processes (not restricted to physical chemistry [29]), although in some cases this description may be somewhat rough. More details dealing with using the equation of autoinhibition and the equations of chemical kinetics in the description of phase transformations and structural changes are available in the previous paper [19] dealing with austenite formation in ferritic nodular cast iron. In addition, other papers [30,31,32] have demonstrated the possible applications of the equations of chemical kinetics for the rough description and modelling of purely physical processes (usually based on diffusion).

Comparing the fits obtained using the two regression functions (3) and (4) shows (see Figure 3) that the fit in the region of increasing curves is comparable, except at 850 °C where the fit obtained using the JMAK equation is much better than that obtained using the equation of autoinhibition. In horizontal parts of curves describing the equilibrium fraction of austenite (martensite), the JMAK equation yielded better fits, because for each temperature (each curve), the equilibrium fraction is calculated as an independent regression parameter, while for the equation of autoinhibition these equilibrium fractions are the functions of the regression parameters connected with all temperatures (all curves)—*quartets* of regression parameters (*k*_∞_, *k*_s__∞_, *ε* and *ε*_s_) or (*T*_min_, *K*, *ε* and *ε*_s_) in the case of using Equation (8a) for *k*(*T*), or *quintets* of regression parameters (*k*_∞_, *k*_s__∞_, *ε*, *ε*_s_ and *a*), or (*T*_min_, *K*, *ε*, *ε*_s_, and *a*) in the case of using Equation (11) for *k*(*T*).

The ability of the autoinhibition model to describe the equilibrium fraction can be discussed for the middle austenitisation temperature (790 °C), where the difference between the measured value, and the fitted value of the equilibrium fraction using the equation of autoinhibition is 2% (or 3 °C if expressed on temperature scale), see Figure 8. Both these error values, i.e., 2% error for volume fraction determination and 3 °C error for temperature regulation and measurement, represent acceptable measurement errors when using standard laboratory equipment and procedures. In addition, the volume fraction of austenite is determined through the volume fraction of martensite, which causes an additional error in fraction determination (the specific volume of martensite is about 4% greater than the specific volume of austenite [33]). Taking this fact into account together with the numbers of regression parameters (JMAK: 14; autoinhibition with modification: 5, see above), the fit obtained using the equation of autoinhibition can be considered to be relatively very successful.

In practice, it can be useful to know the equilibrium volume fraction of martensite *p*_eq_ at given temperature *T* or, vice versa, the austenitisation temperature *T* that is necessary to achieve a desired fraction of martensite *p*_eq_. These values of temperature and martensite fraction can be determined for *p*_eq_ > 0.2 = 20% and *T* > 730 °C using the family of predictive curves (see Figure 4) or the following relationships (inverse of each other):(12)$$p_{\mathrm{eq}}=\exp\left[\frac{\varepsilon -\varepsilon_{s}}{R}\left(\frac{1}{T_{\min}}-\frac{1}{T}\right)\right],$$
(13)$$\frac{1}{T}=\frac{1}{T_{\min}}-\frac{R}{\varepsilon -\varepsilon_{s}}\ln p_{\mathrm{eq}}.$$

Generally, including regions *p*_eq_ < 0.2 = 20% and *T* < 730 °C (1003 K), family of predictive curves in Figure 7 can be used, or Equation (12) is replaced by the more exact relation:(14)$$p_{\mathrm{eq}}=\exp\left[\frac{\varepsilon -\varepsilon_{s}}{R}\left(\frac{1}{T_{\min}}-\frac{1}{T}\right)\right]\tanh[a\;(T-994)],$$

Whose inverse equivalent for *T* (i.e., analogy to Equation (13)) cannot be expressed analytically but Figure 7 is sufficient for determining austenitisation temperature *T*, at which asked martensite fraction *p*_eq_ is formed. If unknown steel is studied, for the determination of the family of predictive curves the values of the martensite volume fraction is necessary to know for at least two austenitisation temperatures and several holding periods. Only if also little martensite fractions are asked (*p*_eq_ < 0.2 = 20%), the results for the third austenitisation temperature close to *Ac*_1_ are necessary as well as the value of *Ac*_1_. Generally, the more experimental results (for various holding periods and various austenitisation temperatures) are available, the more exact values of regression parameters can be determined and the more exact family of predictive curves can be drawn.

The difference between Equations (12) and (14) is presented in Figure 9 where both values of regression parameter *T*_min_ (835.3 and 836.8 °C) for *p*_eq_ = 1 and *Ac*_1_ = 721 °C for *p*_eq_ = 0 are drawn as additional points. Full line representing Equation (14) finishes in zero martensite equilibrium fraction for temperature 721 °C, while dashed line representing Equation (12) reaches zero fraction in the temperature of absolute zero. In addition, the original paper showed the dependence of equilibrium fraction on austenitisation temperature (see Figure 7 in paper [17]). For temperatures 730, 760, 790, 820 and 850 °C the authors used linear fit, which seems to be very successful, but (i) its extrapolation reaches zero martensite equilibrium fraction at 686.5 °C (far below *Ac*_1_ = 721 °C) and (ii) not only at austenitisation temperature of 850 °C, but already at about 837 °C fully martensite structure is obtained if the dwell is sufficiently long. It means that mentioned linear fit is acceptable only in austenitisation temperature region between 730 and 820 °C.

## 8. Conclusions

The results of the regression and modelling of the kinetics of austenite formation in studied dual phase steel enable the authors to formulate the following conclusions:The JMAK equation, which is most frequently used to describe the kinetics of phase transformations, can be used to describe austenitisation, but separately for each temperature. In the case of partial transformation, additional parameters have to be introduced describing the equilibrium volume fraction at each temperature.The equation of autoinhibition was successfully used to describe austenitisation kinetics because the newly formed phase acts against the continuing conversion. This inhibition results from factors including the change in the specific volumes of the initial and final phases, and the chemical liquation of alloying additions.The kinetic equation of autoinhibition together with the Arrhenius equation describes the dependence of conversion of austenitisation on time and temperature with high precision, including for cases of partial conversion, despite the fact that it contains only four parameters (or five parameters if the region close to *Ac*_1_ temperature, i.e., with little martensite fraction, is considered).Successful application of the equation of autoinhibition to the simple description and modelling of the kinetics of austenite formation in dual steels and nodular cast irons leads to the presumption that this approach can be useful for the description of austenitisation in all iron-based alloys, and maybe also for other types of phase transformations. Generally, this approach should be able to achieve at least a rough description of many processes based on two counteracting driving forces with different temperature dependences. This approach describes not only the kinetics of those processes, but also the final equilibrium state (here, the equilibrium volume fraction of austenite), which is outside of the ability of the JMAK equation.A sufficiently extensive set of experimental austenitisation (several values of martensite volume fraction in dual steel for several dwells at a minimum of two or three temperatures), together with the application of the equation of autoinhibition and the Arrhenius equation allows for the construction of predictive curves determining the martensite volume fraction in this steel for arbitrary dwell and temperature combinations. Such construction of predictive curves is not directly possible if the JMAK equation is used for the description of kinetic curves.

## Figures and Tables

**Figure 1 materials-14-01781-f001:**
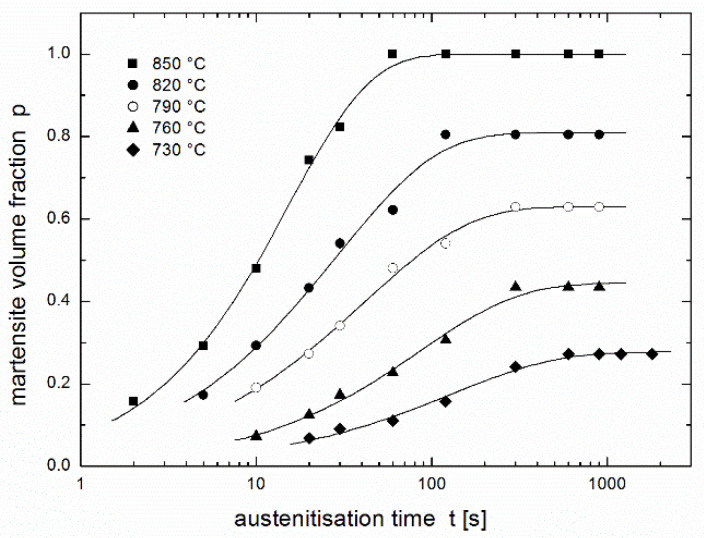
Martensite volume fraction in dual phase steel after austenitisation at various temperatures during various dwells [17] fitted using the JMAK Equation (4) (for each temperature separately).

**Figure 2 materials-14-01781-f002:**
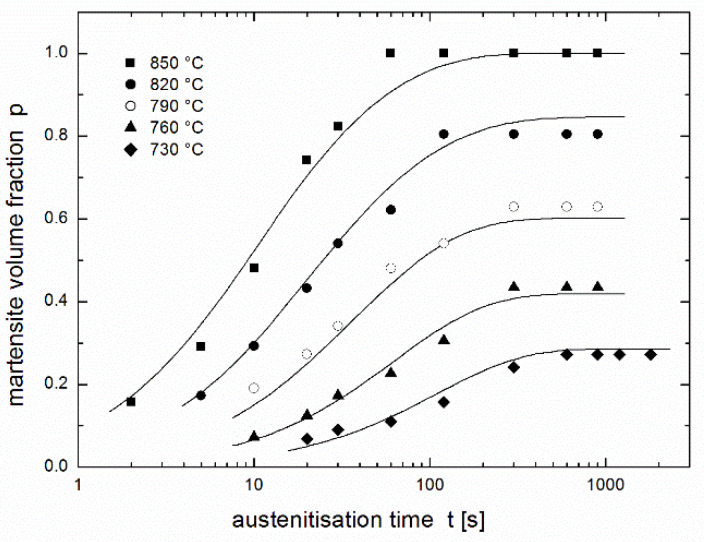
Martensite volume fraction in dual phase steel after austenitisation at various temperatures during various dwells [17] fitted using the equation of autoinhibition (3) and the Arrhenius equation (Equation (2)) (all temperatures were fitted together).

**Figure 3 materials-14-01781-f003:**
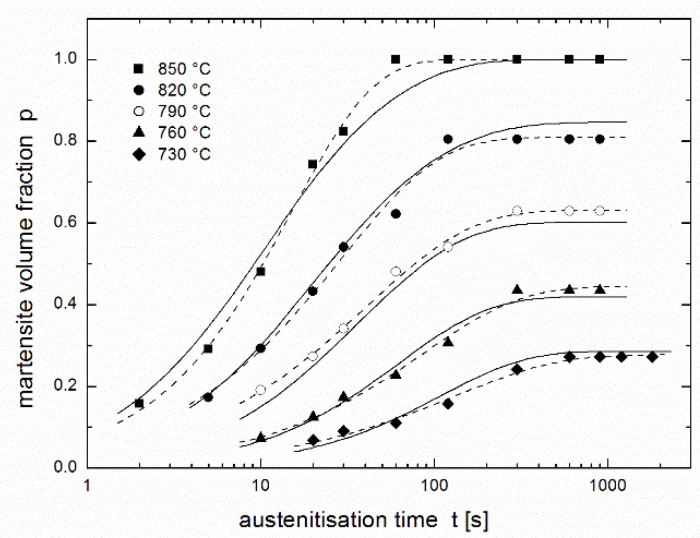
Martensite volume fraction in dual phase steel after austenitisation at various temperatures during various dwells [17]: comparison of fits using the equation of autoinhibition (full lines) and the JMAK equation (dashed lines).

**Figure 4 materials-14-01781-f004:**
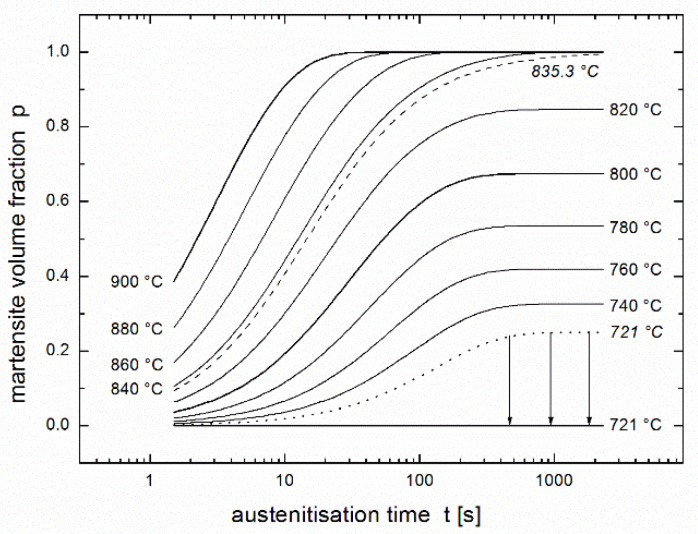
Family of predictive curves for the given temperatures. Prediction is based on the regression using the equation of autoinhibition (3) and the Arrhenius equation (2) or (8), see Figure 2.

**Figure 5 materials-14-01781-f005:**
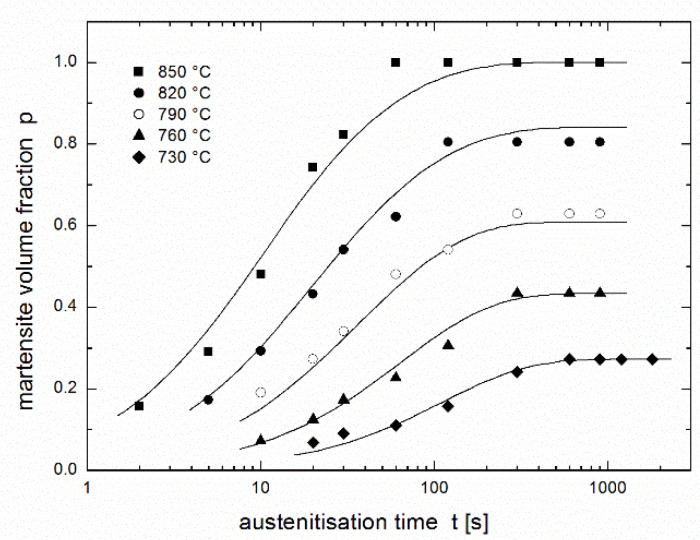
Martensite volume fraction in dual phase steel after austenitisation at various temperatures during various dwells [17] fitted using the equation of autoinhibition (3) and the modified Arrhenius equation (Equation (11)) (all temperatures were fitted together).

**Figure 6 materials-14-01781-f006:**
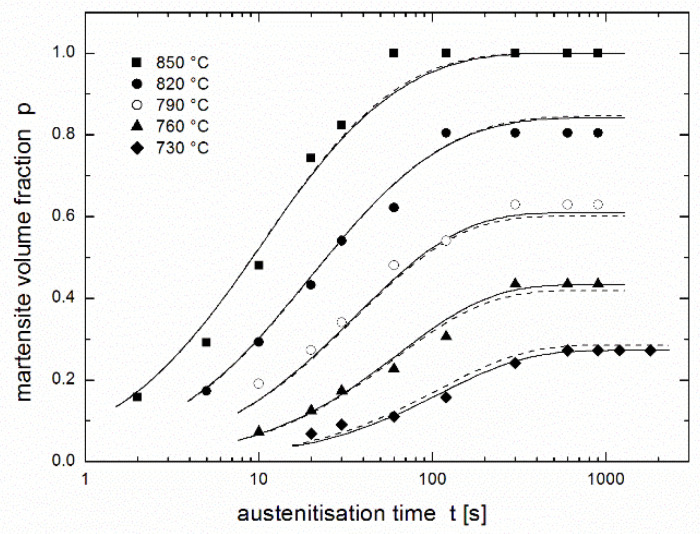
Martensite volume fraction in dual phase steel after austenitisation at various temperatures during various dwells [17]: comparison of fits using the equation of autoinhibition and the Arrhenius equation in a modified form (11) (full lines) and the form of (2) or (8) (dashed lines).

**Figure 7 materials-14-01781-f007:**
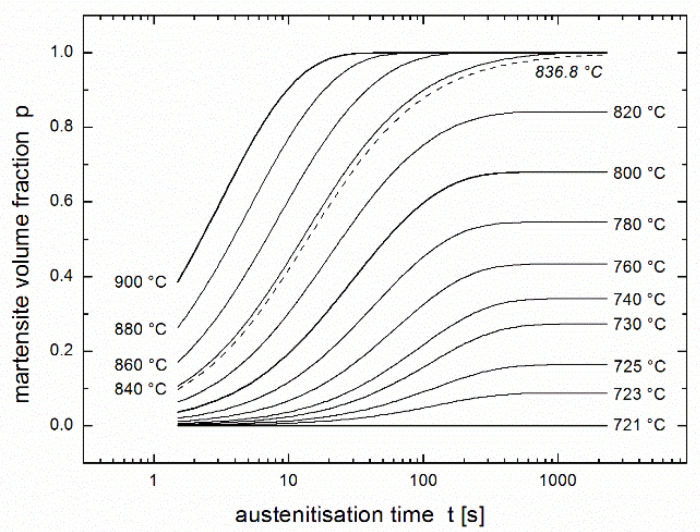
Family of predictive curves for the given temperatures. Prediction is based on the regression using the equation of autoinhibition (3) and the modified Arrhenius equation (11), see Figure 5.

**Figure 8 materials-14-01781-f008:**
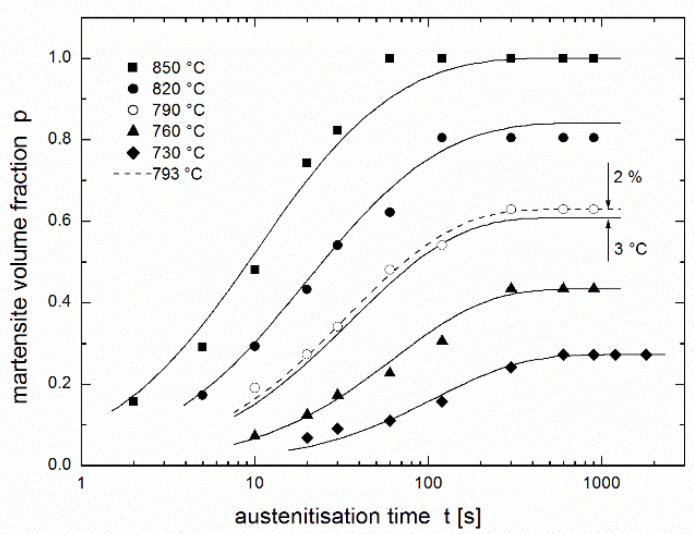
Comparison of the influence of errors in martensite volume fraction determination and austenitisation temperature adjustment on the position of the kinetic curve for 790 °C (the equation of autoinhibition (3) and the modified Arrhenius equation (11) are used).

**Figure 9 materials-14-01781-f009:**
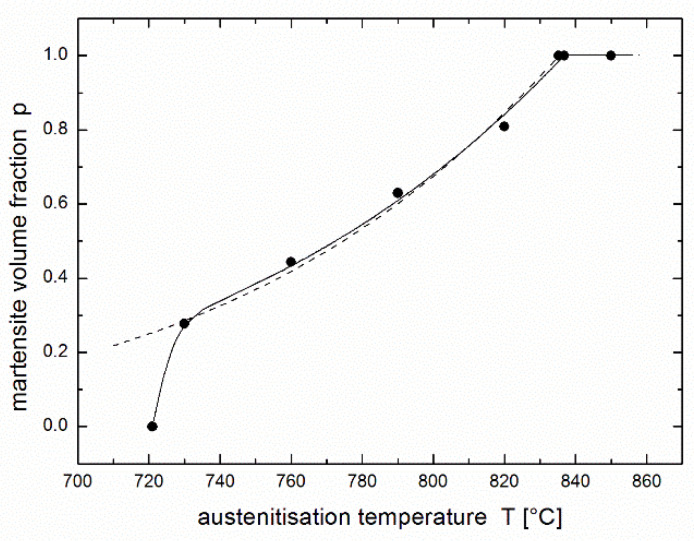
Dependence of equilibrium volume fraction of martensite (austenite) on austenitisation temperature fitted using Equation (12) (dashed line) and Equation (14) (full line). Both temperatures *T*_min_ (835.3 and 836.8 °C) for *p*_eq_ = 1 and *Ac*_1_ = 721 °C for *p*_eq_ = 0 are drawn as additional points.

**Table 1 materials-14-01781-t001:** Different values of the rate constant *k* and kinetic exponent *n* for different forms of the JMAK equation used as regression function: Equation (4) and linearised Equation (5).

Temperature [°C]	730		760		790		820		850	
Regression function	(4)	(5)	(4)	(5)	(4)	(5)	(4)	(5)	(4)	(5)
Rate constant *k* [s^−1^]	0.029	0.033	0.032	0.041	0.066	0.070	0.075	0.077	0.080	0.091
Exponent *n* [1]	0.731	0.712	0.772	0.714	0.731	0.709	0.768	0.775	0.928	0.873

## Data Availability

Data sharing is not applicable to this article.

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
