# Peer review of "Simple and Precise Description of the Transformation Kinetics and Final Structure of Dual Phase Steels"

_materials, 2021, doi:10.3390/ma14071781_

Round 1

Reviewer 1 Report

The manuscript reports an investigation about the kinetics of austenite formation in dual phase steels introducing a modification in the Johnson-Mehl-Avrami-Kolmogorov equation for the case of dual steel with a small content of martensite. This new equation id thr autohinbition equiation and describes the austenite formation.

English is correct and all is correct. O only can suggest minor corrections from the experimental section (although this information is not provided in Asadabad et al 2008):

  • The ASTM E562-83 standard should be cited.
  • You used scanning electron microscopy, then you should indicate which equipment did you use.
  • Conclusions section. You cannot to do reference in conclusions to other papers. Some conclusions are obtained from the results of your research present in other article. Here you should write the conclusions of the research presented in this paper.

Reviewer 2 Report

The content is nice and would be interesting for readers. However, English language should be improved; some sentences should be rephrased

Reviewer 3 Report

Simple and Precise Description of the Transformation Kinetics 2 and Final Structure of Dual Phase Steels

Abstract:

All components are  present;

  • Data are accurate and match text;
  • Well‐written, concise, clear
  • Subject matter is original, press-worthy, of major general interest

Introduction:

Well‐written, concise;

  • Hypothesis and purpose of study are clearly and concisely presented
  • Data are accurate; hypotheses are correctly presented and fully supported by text
  • Current references that will be of interest to readers

Experimental;

Well‐written, concise;

  • Hypothesis and purpose of study are clearly and concisely presented.

Although, description of procedures unclear; would be difficult for others to reproduce the study by reading the article, although with rewriting, the deficit could be remedied

Results:

Logically presented

  • Summarizes important observations

Nevertheless, the Statistical significance of findings not stated, need to be added.

Dissucsion, conclusion:

Statements and conclusions are clearly supported by data and are linked to goals

Minor revision is still needed.

Reviewer 4 Report

Review for

Simple and Precise Description of the Transformation Kinetics and Final Structure of Dual Phase Steels

The level of originality of the paper is high. The method is relevant: Regression using the JMAK equation.

In this paper authors used 28 sources, containing both historical and fundamental works, as well as the latest scientific research on this topic. But the literature review can be improved. The papers discussed many points of this study. Please, add his papers.

Danish, M.S.S., Bhattacharya, A., Stepanova, D., Mikhaylov, A., Grilli, M.L., Khosravy, M., Sengyu, T. (2020). A Systematic Review of Metal Oxide Applications for Energy and Environmental Sustainability. Metals, 10(12), 1604. https://doi.org/10.3390/met10121604

Yumashev, А and Mikhaylov, А. (2020) Development of Polymer Film Coatings with High Adhesion to Steel Alloys and High Wear Resistance. Polymer Composites, 41(7), 2875-2880. https://doi.org/10.1002/pc.25583

The introduction section has benefit from having a clearer structure of what to expect in the paper. Furthermore, the author(s) would benefit from being more concise in their writing, as much of the content was redundant and overemphasized. While it is good practice to assume the reader has no prior knowledge of the content, a topic and/or discussion does not need to be explained over and over again if it is stated both adequately and appropriately once.

Some conclusions contribute to the study of the problem. The author not formulate the problem itself – it makes impossible to analyse the contribution of the paper. The aim or the question of the paper (or even the hypothesis of the author) are formulated.

Overall, it is very clear to grasp understanding of the manuscript and content in its current state. I strongly advise using hypothesis points to articulate and/or express material in scientific writing. Publication of this piece seems likely in any reputable scientific periodical after a correction in the writing of the manuscript.

The paper possesses a proper form of well-structured and readable technical language of the field and represents the expected knowledge of the journal`s readership.

There are minor errors in English, but this does not affect the general nature of the work. The current study brings many new to the existing literature or field. For one, the author(s) seem to have a good grasp of the current literature on their topic area (i.e., recent literature and seminal texts relevant to their study is not cited/referenced).
